# Discovering Potential Compounds for Venous Disease Treatment through Virtual Screening and Network Pharmacology Approach

**DOI:** 10.3390/molecules28247937

**Published:** 2023-12-05

**Authors:** Oscar Salvador Barrera-Vázquez, Juan Luis Escobar-Ramírez, Jacinto Santiago-Mejía, Omar Francisco Carrasco-Ortega, Gil Alfonso Magos-Guerrero

**Affiliations:** Department of Pharmacology, Faculty of Medicine, University National Autonomous of Mexico (UNAM), Mexico City 04510, Mexico; osbarrera6@gmail.com (O.S.B.-V.); re-harakhty@hotmail.com (J.L.E.-R.); samj@unam.mx (J.S.-M.); omarf.carrasco@facmed.unam.mx (O.F.C.-O.)

**Keywords:** phlebotonics, treatment, venous, hypertension, berberine, VD, chemoinformatics, semi-synthetic, natural products

## Abstract

Peripheral venous hypertension has emerged as a prominent characteristic of venous disease (VD). This disease causes lower limb edema due to impaired blood transport in the veins. The phlebotonic drugs in use showed moderate evidence for reducing edema slightly in the lower legs and little or no difference in the quality of life. To enhance the probability of favorable experimental results, a virtual screening procedure was employed to identify molecules with potential therapeutic activity in VD. Compounds obtained from multiple databases, namely AC Discovery, NuBBE, BIOFACQUIM, and InflamNat, were compared with reference compounds. The examination of structural similarity, targets, and signaling pathways in venous diseases allows for the identification of compounds with potential usefulness in VD. The computational tools employed were *rcdk* and *chemminer* from R-Studio and Cytoscape. An extended fingerprint analysis allowed us to obtain 1846 from 41,655 compounds compiled. Only 229 compounds showed pharmacological targets in the PubChem server, of which 84 molecules interacted with the VD network. Because of their descriptors and multi-target capacity, only 18 molecules of 84 were identified as potential candidates for experimental evaluation. We opted to evaluate the berberine compound because of its affordability, and extensive literature support. The experiment showed the proposed activity in an acute venous hypertension model.

## 1. Introduction

Venous disease (VD) is defined as a condition in which the veins cannot transport blood unidirectionally to the heart with a flow adapted to the needs of tissue drainage, temperature regulation, and hemodynamic reserve, regardless of their position and activity. The first manifestation of VD is increased venous pressure (venous hypertension or high blood pressure in the veins) with or without reflux [1]. In the United States, it affects over 25 million adults and over 6 million have advanced disease. The total cost of medical care is estimated at more than USD 3 billion per year [2]. In Latin America, it is estimated that the prevalence of VD is 68.11% [3], and in Mexico, the incidence rate in 2022 was 142.27/100,000 inhabitants [4].

The treatments of choice for VD are surgery, sclerotherapy, and mechanical compression. However, pharmacological treatment with drugs referred to as phlebotonics is frequently used due to their easy administration. In the Anatomical Therapeutic Chemical (ATC) system, phlebotonics are categorized as vasoprotective agents, specifically within the group of capillary stabilizing agents (ATC 2022) [5]. Most of these agents are derived from natural flavonoids extracted from plants [6].

Although phlebotonics are known as vasoactive drugs, their mechanism of action is not scientifically well established despite the lack of studies examining their pharmacological and clinical properties. These drugs have been found to impact macrocirculation, such as enhancing venous tone [7], as well as microcirculatory parameters such as capillary hyper-permeability [8]. Controversy exists regarding the clinical relevance of the efficacy and benefit–risk balance of phlebotonics. A meta-analysis study suggests a slight reduction in leg swelling, but it may not significantly affect quality of life or ulcer healing. Phlebotonics probably increase adverse events such as gastrointestinal disorders [6]. The therapeutic benefits of phlebotonics may not be fully realized in the VD because of their limitations and mechanisms of action. 

Computer-aided drug design (CADD) is a discipline that uses a range of chemical-molecular and quantum methodologies. CADD allows us to investigate, develop, and synthesize medicinal chemical compounds [9]. One computational approach used in computer-aided drug design involves the application of chemoinformatics. This discipline specifically concentrates on extracting, processing, and extrapolating significant data from chemical structures [9]. Chemoinformatics is used to compare drugs, natural products (NPs), and semi-synthetic molecules (SMSs) to find potential candidates for pre-clinical research. 

CADD has been promoted by various disciplines, such as chemoinformatics and network pharmacology, which were employed in the present investigation to find new drugs for VD treatment. CADD has proven to be effective in finding new candidates from NPs and SMSs that succeed better and have fewer side effects [10]. 

We tested a candidate found through virtual screening to address increased venous pressure, which is the first manifestation of VD. This evaluation was performed in a rat model of acute venous hypertension. 

## 2. Results 

### 2.1. Creation of a Reference Compounds Dataset

During the first phase of our research, MEDLINE, EMBASE, and Scopus servers enabled us to generate a dataset of reference compounds with phlebotonic, anti-inflammatory, and antioxidant properties. We found 145 reference compounds. Osiris DataWarrior software V5.2.1 and SwissADME server were used to determine their one-dimensional (1D) and two-dimensional (2D) molecular descriptors. Through the use of hierarchical clustering (K-means) and principal component analysis (PCA) via RStudio, an analysis was conducted on the molecular descriptors to determine the most significant reference compounds obtained from the literature (Figure 1). Appendix A contains the PCA results. The molecular descriptors include Fragments, Rotatable Bonds, Veber #violations, Molecular Flexibility, Fraction Csp3 Stereo Centers, sp3-Atoms, Saturated Rings, Non-Aromatic Rings, Molecular Complexity, MR, Total Surface Area, #Heavy atoms, Non-H Atoms, Molweight, Monoisotopic Mass, Total Molweight, Synthetic Accessibility, Hetero-Rings, Small Rings, Rings Closures, H-Donors, Polar Surface Area, TPSA, H-Acceptors, Non-C/H Atoms, Electronegative Atoms, and Muegge #violations. Cluster one showed the greatest similarity between the following 23 molecules: escin, crocin, echinomycin, cyclosporine A, amphotericin B, everolimus, rapamycin, chetomin, astragaloside IV, 20(R)-ginsenoside Rh2, epigallocatechin 3-gallate, troxerutin, O-(beta-hydroxyethyl)-rutoside, salvianolic acid B, keracyanin chloride, rutin, hidrosmin, diosmin, hesperidin, linarin, isorhoifolin, naringin, and betanin. Because of their great similarity in their molecular descriptors, these compounds are marked in green (Figure 1). The outstanding compounds were classified as reference compounds for the rest of the study and comprise various chemical families, such as saponins, carotenoids, cyclic peptides, macrolides, and alkaloids. Many flavonoids, including flavonoid glycosides, flavones, flavanols (catechins), flavonols, and anthocyanins, are present. Escin, diosmin, and hesperidin are used as first-line treatments for VD [11]. Table 1 presents a summary of the characteristics of the compounds in cluster one, including their activity and K-means coefficient. Escin, crocin, echinomycin, astragaloside IV, epigallocatechin 3-gallate, cyclosporin A, chetomin, and troxerutin were the most similar because of their high K-means coefficient. 

**Table 1 molecules-28-07937-t001:** Main characteristics of the molecules of cluster one.

Reference Compounds	IUPAC Name	PubChem CID	Pharmacological Activity	K-Means Coefficient	References
Escin	(2*S*,3*S*,4*S*,5*R*,6*R*)-6-[[(4*S*,6*bS*,8*R*,9*R*)-9-acetyloxy-8-hydroxy-4,8*a*-bis(hydroxymethyl)-4,6*a*,6*b*,11,11,14*b*-hexamethyl-10-[(*E*)-2-methylbut-2-enoyl]oxy-1,2,3,4*a*,5,6,7,8,9,10,12,12*a*,14,14*a*-tetradecahydropicen-3-yl]oxy]-4-hydroxy-3-[(2*S*,3*R*,4*S*,5*S*,6*R*)-3,4,5-trihydroxy-6-(hydroxymethyl)oxan-2-yl]oxy-5-[(2*R*,3*R*,4*S*,5*S*,6*R*)-3,4,5-trihydroxy-6-(hydroxymethyl)oxan-2-yl]oxyoxane-2-carboxylic acid	6476031	Anti-edematous, anti-inflammatory, and venotonic agent	2.03	[12]
Crocin	bis[(2*S*,3*R*,4*S*,5*S*,6*R*)-3,4,5-trihydroxy-6-[[(2*R*,3*R*,4*S*,5*S*,6*R*)-3,4,5-trihydroxy-6-(hydroxymethyl)oxan-2-yl]oxymethyl]oxan-2-yl] (2*E*,4*E*,6*E*,8*E*,10*E*,12*E*,14*E*)-2,6,11,15-tetramethylhexadeca-2,4,6,8,10,12,14-heptaenedioate	5281233	Inhibitor inhibits STAT3 activation induced by IL-6	1.50	[13]
Echinomycin	*N*-[2,4,12,15,17,25-hexamethyl-27-methylsulfanyl-3,6,10,13,16,19,23,26-octaoxo-11,24-di(propan-2-yl)-20-(quinoxaline-2-carbonylamino)-9,22-dioxa-28-thia-2,5,12,15,18,25-hexazabicyclo[12.12.3]nonacosan-7-yl]quinoxaline-2-carboxamide	3197	Anti-cancer agent and inhibitor of HIF	1.39	[14]
Cyclosporin A	(3*S*,6*S*,9*S*,12*R*,15*S*,18*S*,21*S*,24*S*,30*S*,33*S*)-30-ethyl-33-[(*E*,1*R*,2*R*)-1-hydroxy-2-methylhex-4-enyl]-1,4,7,10,12,15,19,25,28-nonamethyl-6,9,18,24-tetrakis(2-methylpropyl)-3,21-di(propan-2-yl)-1,4,7,10,13,16,19,22,25,28,31-undecazacyclotritriacontane-2,5,8,11,14,17,20,23,26,29,32-undecone	5284373	Immunosuppressant agent and inhibitor of TCR signaling via NFAT-independent	1.20	[15]
Amphotericin B	(1*R*,3*S*,5*R*,6*R*,9*R*,11*R*,15*S*,16*R*,17*R*,18*S*,19*E*,21*E*,23*E*,25*E*,27*E*,29*E*,31*E*,33*R*,35*S*,36*R*,37*S*)-33-[(2*R*,3*S*,4*S*,5*S*,6*R*)-4-amino-3,5-dihydroxy-6-methyloxan-2-yl]oxy-1,3,5,6,9,11,17,37-octahydroxy-15,16,18-trimethyl-13-oxo-14,39-dioxabicyclo[33.3.1]nonatriaconta-19,21,23,25,27,29,31-heptaene-36-carboxylic acid	5280965	Antibiotic agent for the treatment of life-threatening fungal infections and modulator of the immune system	0.97	[16]
Everolimus	(1*R*,9*S*,12*S*,15*R*,16*E*,18*R*,19*R*,21*R*,23*S*,24*E*,26*E*,28*E*,30*S*,32*S*,35*R*)-1,18-dihydroxy-12-[(2*R*)-1-[(1*S*,3*R*,4*R*)-4-(2-hydroxyethoxy)-3-methoxycyclohexyl]propan-2-yl]-19,30-dimethoxy-15,17,21,23,29,35-hexamethyl-11,36-dioxa-4-azatricyclo[30.3.1.0^4,9^]hexatriaconta-16,24,26,28-tetraene-2,3,10,14,20-pentone	6442177	Anticancer agent and mTOR inhibitor	0.99	[17]
Rapamycin	(1*R*,9*S*,12*S*,15*R*,16*E*,18*R*,19*R*,21*R*,23*S*,24*E*,26*E*,28*E*,30*S*,32*S*,35*R*)-1,18-dihydroxy-12-[(2*R*)-1-[(1*S*,3*R*,4*R*)-4-hydroxy-3-methoxycyclohexyl]propan-2-yl]-19,30-dimethoxy-15,17,21,23,29,35-hexamethyl-11,36-dioxa-4-azatricyclo[30.3.1.0^4,9^]hexatriaconta-16,24,26,28-tetraene-2,3,10,14,20-pentone	5284616	Inhibitor of mTOR complex 1 (mTORC1), which phosphorylates substrates including S6 kinase 1 (S6K1), eIF4E-binding protein 1 (4E-BP1), transcription factor EB (TFEB), unc-51-like autophagy-activating kinase 1 (Ulk1), and growth factor receptor-bound protein 10 (GRB-10)	0.91	[18]
Chetomin	14-(hydroxymethyl)-3-[3-[[4-(hydroxymethyl)-5,7-dimethyl-6,8-dioxo-2,3-dithia-5,7-diazabicyclo[2.2.2]octan-1-yl]methyl]indol-1-yl]-18-methyl-15,16-dithia-10,12,18-triazapentacyclo[12.2.2.0^1,12^.0^3,11^.0^4,9^]octadeca-4,6,8-triene-13,17-dione	10417379	Inhibitor of HIF-1α/p300 interaction, and antitumor agent	1.16	[19]
Astragaloside IV	(2*R*,3*R*,4*S*,5*S*,6*R*)-2-[[(1*S*,3*R*,6*S*,8*R*,9*S*,11*S*,12*S*,14*S*,15*R*,16*R*)-14-hydroxy-15-[(2*R*,5*S*)-5-(2-hydroxypropan-2-yl)-2-methyloxolan-2-yl]-7,7,12,16-tetramethyl-6-[(2*S*,3*R*,4*S*,5*R*)-3,4,5-trihydroxyoxan-2-yl]oxy-9-pentacyclo[9.7.0.0^1,3^.0^3,8^.0^12,16^]octadecanyl]oxy]-6-(hydroxymethyl)oxane-3,4,5-triol	13943297	Anti-inflammatory agent and inhibitor of NF-kappaB activation and adhesion molecule expression	1.29	[20]
20(R)-ginsenoside Rh2	(2*R*,3*R*,4*S*,5*S*,6*R*)-2-[[(3*S*,5*R*,8*R*,9*R*,10*R*,12*R*,13*R*,14*R*,17*S*)-12-hydroxy-17-[(2*R*)-2-hydroxy-6-methylhept-5-en-2-yl]-4,4,8,10,14-pentamethyl-2,3,5,6,7,9,11,12,13,15,16,17-dodecahydro-1*H*-cyclopenta[a]phenanthren-3-yl]oxy]-6-(hydroxymethyl)oxane-3,4,5-triol	54580480	Antitumoral agent	0.63	[21]
Epigallocatechin 3-gallate	(2*S*,3*S*,4*S*,5*R*,6*S*)-6-[2,6-dihydroxy-4-[[(2*R*,3*R*)-5-hydroxy-7-[(2*S*,3*R*,4*S*,5*S*,6*R*)-3,4,5-trihydroxy-6-(hydroxymethyl)oxan-2-yl]oxy-2-(3,4,5-trihydroxyphenyl)-3,4-dihydro-2*H*-chromen-3-yl]oxycarbonyl]phenoxy]-3,4,5-trihydroxyoxane-2-carboxylic acid	102025303	Anti-carcinogen, anti-tumorigenesis, anti-proliferation, anti-angiogenesis, and antioxidant agent, cell death inductor	1.27	[22]
Troxerutin	2-[3,4-bis(2-hydroxyethoxy)phenyl]-5-hydroxy-7-(2-hydroxyethoxy)-3-[(2*S*,3*R*,4*S*,5*S*,6*R*)-3,4,5-trihydroxy-6-[[(2*R*,3*R*,4*R*,5*R*,6*S*)-3,4,5-trihydroxy-6-methyloxan-2-yl]oxymethyl]oxan-2-yl]oxychromen-4-one	5486699	Radioprotective and antioxidant agent	1.12	[23]
O-(beta-hydroxyethyl)-rutoside	2-(3,4-dihydroxyphenyl)-5-hydroxy-7-(2-hydroxyethoxy)-3-[(2*S*,3*R*,4*S*,5*S*,6*R*)-3,4,5-trihydroxy-6-[[(2*R*,3*R*,4*R*,5*R*,6*S*)-3,4,5-trihydroxy-6-methyloxan-2-yl]oxymethyl]oxan-2-yl]oxychromen-4-one	9852585	An agent used as a treatment for disorders of the venous and microcirculatory systems	0.95	[24]
Salvianolic acid B	(2*R*)-2-[(*E*)-3-[(2*S*,3*S*)-3-[(1*R*)-1-carboxy-2-(3,4-dihydroxyphenyl)ethoxy]carbonyl-2-(3,4-dihydroxyphenyl)-7-hydroxy-2,3-dihydro-1-benzofuran-4-yl]prop-2-enoyl]oxy-3-(3,4-dihydroxyphenyl)propanoic acid	6451084	Anti-inflammatory and antioxidant agent	0.98	[25]
Keracyanin chloride	(2*R*,3*R*,4*R*,5*R*,6*S*)-2-[[(2*R*,3*S*,4*S*,5*R*,6*S*)-6-[2-(3,4-dihydroxyphenyl)-5,7-dihydroxychromenylium-3-yl]oxy-3,4,5-trihydroxyoxan-2-yl]methoxy]-6-methyloxane-3,4,5-triol chloride	29231	Anti-inflammatory and antioxidant agent	0.85	[26]
Rutin	2-(3,4-dihydroxyphenyl)-5,7-dihydroxy-3-[(2*S*,3*R*,4*S*,5*S*,6*R*)-3,4,5-trihydroxy-6-[[(2*R*,3*R*,4*R*,5*R*,6*S*)-3,4,5-trihydroxy-6-methyloxan-2-yl]oxymethyl]oxan-2-yl]oxychromen-4-one	5280805	Anticancer, antidiabetic, antimicrobial, anticoagulant, antioxidant, cytoprotective, vasoprotective, anticarcinogenic, neuroprotective and cardioprotective agent	0.85	[27]
Hidrosmin	5-(2-hydroxyethoxy)-2-(3-hydroxy-4-methoxyphenyl)-7-[(2*S*,3*R*,4*S*,5*S*,6*R*)-3,4,5-trihydroxy-6-[[(2*R*,3*R*,4*R*,5*R*,6*S*)-3,4,5-trihydroxy-6-methyloxan-2-yl]oxymethyl]oxan-2-yl]oxychromen-4-one	3087722	Venoactive agent and post-thrombotic syndrome protector	0.88	[28]
Diosmin	5-hydroxy-2-(3-hydroxy-4-methoxyphenyl)-7-[(2*S*,3*R*,4*S*,5*S*,6*R*)-3,4,5-trihydroxy-6-[[(2*R*,3*R*,4*R*,5*R*,6*S*)-3,4,5-trihydroxy-6-methyloxan-2-yl]oxymethyl]oxan-2-yl]oxychromen-4-one	5281613	Agent for treatment of chronic venous insufficiency and varicose veins, with antioxidant, anticancer activities	0.82	[29,30]
Hesperidin	(2*S*)-5-hydroxy-2-(3-hydroxy-4-methoxyphenyl)-7-[(2*S*,3*R*,4*S*,5*S*,6*R*)-3,4,5-trihydroxy-6-[[(2*R*,3*R*,4*R*,5*R*,6*S*)-3,4,5-trihydroxy-6-methyloxan-2-yl]oxymethyl]oxan-2-yl]oxy-2,3-dihydrochromen-4-one	53477767	Antioxidant, neuroprotective, and anti-inflammatory agent	0.82	[31]
Linarin	5-hydroxy-2-(4-methoxyphenyl)-7-[(2*S*,3*R*,4*S*,5*S*,6*R*)-3,4,5-trihydroxy-6-[[(2*R*,3*R*,4*R*,5*R*,6*S*)-3,4,5-trihydroxy-6-methyloxan-2-yl]oxymethyl]oxan-2-yl]oxychromen-4-one	5317025	Antioxidant and anti-inflammatory agents	0.76	[32]
Isorhoifolin	5-hydroxy-2-(4-hydroxyphenyl)-7-[(2*S*,3*R*,4*S*,5*S*,6*R*)-3,4,5-trihydroxy-6-[[(2*R*,3*R*,4*R*,5*R*,6*S*)-3,4,5-trihydroxy-6-methyloxan-2-yl]oxymethyl]oxan-2-yl]oxychromen-4-one	9851181	Antioxidant agent	0.75	[33]
Naringin	(2*S*)-7-[(2*S*,3*R*,4*S*,5*S*,6*R*)-4,5-dihydroxy-6-(hydroxymethyl)-3-[(2*S*,3*R*,4*R*,5*R*,6*S*)-3,4,5-trihydroxy-6-methyloxan-2-yl]oxyoxan-2-yl]oxy-5-hydroxy-2-(4-hydroxyphenyl)-2,3-dihydrochromen-4-one	442428	Antioxidant, antitumor, antiviral, antibacterial, anti-inflammatory, anti-adipogenic, and cardioprotective agent	0.72	[34]
Betanin	1-[(2*E*)-2-(2,6-dicarboxy-2,3-dihydro-1*H*-pyridin-4-ylidene)ethylidene]-6-hydroxy-5-[(2*S*,3*R*,4*S*,5*S*,6*R*)-3,4,5-trihydroxy-6-(hydroxymethyl)oxan-2-yl]oxy-2,3-dihydroindol-1-ium-2-carboxylate	12300103	Antioxidant and anti-inflammatory agents	0.46	[35]

### 2.2. Selection of Drug-like NPs and SMSs Based on QED

We searched candidates compounds with potential usefulness in VD from NP and SMSs molecules collected from several databases. At the end of the search, 41,655 molecules derived from InflamNat (657 molecules), BIOFACQUIM (422 molecules), NuBBE (155 molecules), and AC Discovery (7320 NP molecules and 33,063 SMS molecules) were obtained. The molecular descriptors of 41,655 molecules and 145 reference molecules were analyzed to examine their distribution regarding the QED (Quantitative Estimate of Druglikeness) index. These results are shown in Figure 2 and Figure 3, where it is observed that the values of molecular descriptors for NPs, SMSs, and reference compounds are distributed within a similar range. This tendency in the distribution is attributed to the fact that most reference compounds are derived from natural sources.

At the end of QED analysis, molecules with a score greater than 0.5 were considered drug-like by their potential use of being administered orally. From 41,655 molecules, 27,868 molecules passed the rule (QED > 0.5), of which 2964 were NP molecules (10.1%) and 24,904 were SMS molecules (96.51%). The total number of molecules that violated the rule (QED 0.5) was 12,670 molecules (4511 were NP molecules and 8159 were SMS molecules).

To analyze and remove compounds that have non-desirable effects, we used the Osiris Data Warrior software V5.2.1. This analysis identified 26,530 molecules (1626 NP molecules and 24,904 SMS molecules) free from potential tumorigenic and deleterious functions, including in the reproductive system. The chemical space of our molecules obtained using three-dimensional principal component analysis (3D-PCA) allowed us to identify drug-like NPs and SMSs that were similar to our reference compounds. At the end of the analysis, 1953 similar molecules (1293 NPs and 660 SMSs) were identified by their chemical space (Figure 4). 

### 2.3. Selection of NPs and SMSs Structurally Similar to Reference Compounds

To further analyze the obtained result via 3D-PCA, we performed a fingerprint analysis. Of the 1953 drug-like molecules obtained via 3D-PCA, 1846 were found in PubChem because this server contains various types of structure searches in 2D, which are important for fingerprint analysis. To achieve this aim, a comparison was made between the chemical structure of 1846 drug-like molecules and the 23 reference compounds obtained from cluster 1. Figure 5 presents an illustrative example of fingerprint analysis using the reference compound diosmin. The elbow method shows that the optimal number of clusters for clusterization is three (Figure 5A). Figure 5B shows the comparison of fingerprints between the 1846 drug-like molecules and the reference compounds through the Tanimoto coefficient and Ward’s method clustering. Figure 5C,D illustrate the Dunn index and silhouette coefficient (Sc), which confirm the clusters obtained using Ward’s method for the 1846 drug-like molecules. The number of drug-like molecules similar to each one from the 23 reference compounds is shown in Table 2. The highest Sc value is found in the cluster that contains the phlebotonic reference compound. This result suggests a strong similarity between 1846 drug-like molecules and 23 reference compounds. Finally, the mean of the Sc of the clusterization performed with the 23 reference compounds was 0.35. Appendix A shows the complete analysis for each of the 23 reference compounds. Appendix A show the clusters of those compounds that were structurally similar to the reference compounds. This information is shown in these sections since it is difficult to present it clearly in Figure 5 because of the large amount of data.

### 2.4. Network Analysis to Obtain the Best Molecules with Potential Usefulness in VD through Their Multi-Target Capacity

An important requirement for the construction of structural networks is that analyzed molecules interact with pharmacological targets. From the 1846 molecules obtained from fingerprint analysis, only 229 showed pharmacological targets on the PubChem server. To enhance compound interaction with the VD networks, a macro instruction was used to match all the involved genes. At the end of the macro, only 84 out of 229 molecules interacted directly with the genes involved in the VD networks (Table 3). A compound–target network studied the interactions between the genes and molecules with potential usefulness in VD. Our compound–target network (Figure 6) for VD comprised 1430 nodes and 2232 edges. At the end of the network analysis, 23 molecules were identified with the best multi-target capacity. However, only 18 molecules have the largest number of targets involved in more than one pathway of the VD network. So, we identified these 18 molecules with a multi-target and multi-pathway capacity (Figure 6). 

The 18 potential phlebotonic molecules are represented in Figure 6 as yellow nodes and their pharmacological targets as orange nodes. These potential candidates are (*E*)-3-phenyl-*N*-(2-phenylethyl)prop-2-enamide; 3-hydroxy-1-(4-hydroxyphenyl)propan-1-one; 2-(4-methoxyphenyl)-2,3-dihydrochromen-4-one; 5-hydroxy-7-methoxy-3-(4-methoxyphenyl)chromen-4-one; 7-hydroxy-2-phenylchromen-4-one; 5,7-dihydroxy-2-(4-methoxyphenyl)chromen-4-one; 2-hydroxy-6-(4-hydroxy-2-methoxy-6-methoxycarbonylphenoxy)-4-methylbenzoic acid; 16,17-dimethoxy-5,7-dioxa-13-azoniapentacyclo[11.8.0.0^2,10^.0^4,8^.0^15,20^]henicosa-1(13),2,4(8),9,14,16,18,20-octaene chloride; 5,7-dihydroxy-3-(4-methoxyphenyl)chromen-4-one; (*E*)-*N*-[(4-hydroxy-3-methoxyphenyl)methyl]-8-methylnon-6-enamide; (*E*)-1-(2,4-dihydroxy-6-methoxyphenyl)-3-phenylprop-2-en-1-one; 5,7-dihydroxy-2-phenylchromen-4-one; 7-hydroxy-3-(4-methoxyphenyl)chromen-4-one; 9-methoxy-2,2-dimethyl-6*H*-pyrano[3,2-c]quinolin-5-one; (2*S*)-5,7-dihydroxy-2-(4-hydroxyphenyl)-2,3-dihydrochromen-4-one; 1-[(3,4-dimethoxyphenyl)methyl]-6,7-dimethoxyisoquinoline;hydrochloride; 3-(4-hydroxyphenyl)-1-(2,4,6-trihydroxyphenyl)propan-1-one; and (2*Z*)-2-[(3,4-dihydroxyphenyl)methylidene]-6-hydroxy-1-benzofuran-3-one. Table 4 summarizes the characteristics of compounds with potential usefulness in VD derived from the compound–target network.

**Table 4 molecules-28-07937-t004:** Main characteristics of the best candidate compounds with usefulness in VD obtained via the multi-target capacity.

Compounds	IUPAC Name	Structure	Source	Pharmacological Activity	Number of Targets	References
N-Phenethylcinnamamide CID 795855	(*E*)-3-phenyl-*N*-(2-phenylethyl)prop-2-enamide	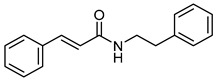	*Acmella radicans*	Inhibitor agent of prostaglandin and leukotriene biosyntheses	4	[36,37]
3,4′-Dihydroxypropiophenone CID 638759	3-hydroxy-1-(4-hydroxyphenyl)propan-1-one	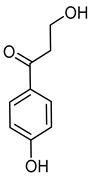	The roots of the *Carissa edulis*	AChE inhibitor, cytotoxic and antioxidant agent.	7	[38]
4′-Methoxyflavanone CID 102928	2-(4-methoxyphenyl)-2,3-dihydrochromen-4-one	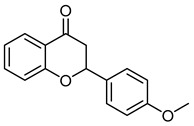	Isolated from *Isaria fumosorosea*	Inhibitor agent of glycation and aldose reductase activity	4	[39]
7,4′-dimethoxy-5-hydroxy isoflavone CID 5386259	5-hydroxy-7-methoxy-3-(4-methoxyphenyl)chromen-4-one	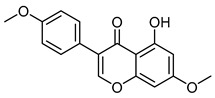	Isolated from the roots of *Lotus polyphyllos*	Anticancer agent	6	[40,41]
7-hydroxyflavone CID 5281894	7-hydroxy-2-phenylchromen-4-one	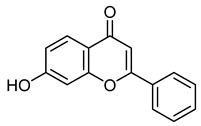	Isolated from *M. indica*	Vasorelaxant and anti-inflammatory agent	5	[42,43,44]
Acacetin CID 5280442	5,7-dihydroxy-2-(4-methoxyphenyl)chromen-4-one	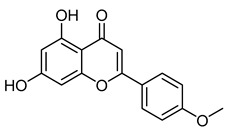	*Robinia pseudoacacia*	Antiviral agent	19	[45]
Asterric acid CID 3080568	2-hydroxy-6-(4-hydroxy-2-methoxy-6-methoxycarbonylphenoxy)-4-methylbenzoic acid	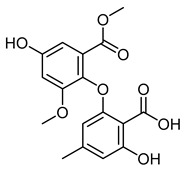	*Aspergillus terreus*	Antibiotic agent	4	[46]
Berberine chloride CID 12456	16,17-dimethoxy-5,7-dioxa-13-azoniapentacyclo[11.8.0.0^2,10^.0^4,8^.0^15,20^]henicosa-1(13),2,4(8),9,14,16,18,20-octaene chloride	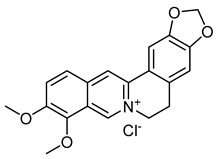	*Rhizoma coptidis*	Antibacterial and anti-inflammatory agents	4	[47]
Biochanin A CID 5280373	5,7-dihydroxy-3-(4-methoxyphenyl)chromen-4-one	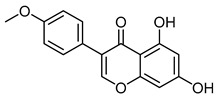	Red clover, soy, alfalfa sprouts, peanuts, chickpea (*Cicer arietinum*), and other legumes	Antiviral agent	9	[48]
Capsaicin CID 1548943	(*E*)-*N*-[(4-hydroxy-3-methoxyphenyl)methyl]-8-methylnon-6-enamide	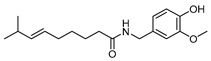	Chili peppers	The agent involved in the treatment of obesity, diabetes, cardiovascular conditions, cancer, airway diseases, itch, gastric, and urological disorders	4	[49]
Cardamonin CID 641785	(*E*)-1-(2,4-dihydroxy-6-methoxyphenyl)-3-phenylprop-2-en-1-one	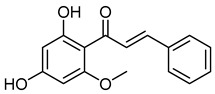	*Alpinia blepharocalyx, Alpinia gagnepainii, Alpinia conchigera, Alpinia hainanensis, Alpinia malaccensis, Alpinia mutica, Alpinia pricei, Alpinia rafflesiana, Alpinia speciosa, Amomum subulatum, Artemisia absinthium, Boesenbergia pandurata, Boesenbergia rotunda, Carya cathayensis, Cedrelopsis grevei, Combretum apiculatum, Comptonia peregrina, Desmos cochinchinensis, Elettaria cardamomum, Helichrysum forskahlii, Kaempferia parviflora, Morella pensylvanica, Piper dilatatum, Piper hispidum, Polygonum ferrugineum, Polygonum lapathifolium, Polygonum persicaria, Populus fremontii, Populus × euramericana*, a hybrid between *Populus deltoides* and *Populus nigra, Syzygium samarangense, Vitex leptobotrys*, and *Woodsia scopulina.*	Anti-inflammatory, antineoplastic, and antioxidant agent	4	[50]
Chrysin CID 5281607	5,7-dihydroxy-2-phenylchromen-4-one	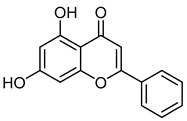	Isolated from *Pyrus pashia* fruit	Vasorelaxant agent	18	[42,51]
Formononetin CID 5280378	7-hydroxy-3-(4-methoxyphenyl)chromen-4-one	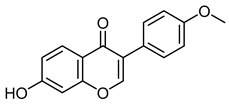	Red clover	Anti-cancer agent	4	[52]
Haplamine CID 648601	9-methoxy-2,2-dimethyl-6*H*-pyrano[3,2-c]quinolin-5-one	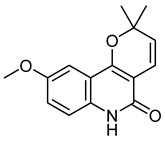	Isolated from *Haplophyllum perforatu*	Anti-cancer agent	9	[53]
Naringenin CID 439246	(2*S*)-5,7-dihydroxy-2-(4-hydroxyphenyl)-2,3-dihydrochromen-4-one	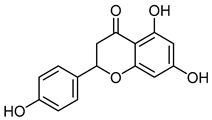	Citrus fruits, bergamot, tomatoes, and other fruits	Anti-Hepatitis C virus, antiaging, anti-Alzheimer, antiasthma, anticancer, anti-Chikungunya virus, anticonvulsant, anti-dengue virus, antidiabetic, anti-Edwardsiellosis, anti-hyperlipidemic, anti-inflammatory, antimicrobial, antioxidant, antiplatelet, anti-stroke damage, cardioprotective, chronic kidney disease, expectorant, eye-protective, fertility, immunomodulatory, laxative, hepatoprotective, pregnancy, radioprotective, and weight-loss agent	4	[34]
Papaverine hydrochloride CID 6084	1-[(3,4-dimethoxyphenyl)methyl]-6,7-dimethoxyisoquinoline;hydrochloride	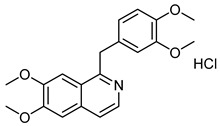	*Papaver somniferum* (Opium poppy)	Vasodilator agent	4	[54]
Phloretin CID 4788	3-(4-hydroxyphenyl)-1-(2,4,6-trihydroxyphenyl)propan-1-one	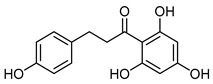	Apple	Antioxidative, anti-inflammatory, anti-microbial, anti-allergic, anticarcinogenic, anti-thrombotic, and hepatoprotective agent	4	[55]
Sulfuretin CID 5281295	(2*Z*)-2-[(3,4-dihydroxyphenyl)methylidene]-6-hydroxy-1-benzofuran-3-one	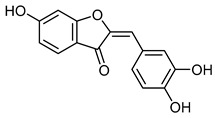	The stem bark of *Albizzia julibrissin* and heartwood of *Rhus verniciflua*	Adipogenesis inhibitor	7	[56]

### 2.5. Selection of Berberine for Experimental Validation

Once we found the most promising candidates with potential usefulness in the VD (18 compounds), we selected the following seven compounds as the most suitable to be tested experimentally: 5-hydroxy-7-methoxy-3-(4-methoxyphenyl)chromen-4-one; 5,7-dihydroxy-2-(4-methoxyphenyl)chromen-4-one; 16,17-dimethoxy-5,7-dioxa-13-azoniapentacyclo[11.8.0.0^2,10^.0^4,8^.0^15,20^]henicosa-1(13),2,4(8),9,14,16,18,20-octaene chloride; 5,7-dihydroxy-3-(4-methoxyphenyl)chromen-4-one; 5,7-dihydroxy-2-phenylchromen-4-one; 7-hydroxy-3-(4-methoxyphenyl)chromen-4-one; and (2Z)-2-[(3,4-dihydroxyphenyl)methylidene]-6-hydroxy-1-benzofuran-3-one. Because of the availability, accessibility, administration, and literature background, 16,17-dimethoxy-5,7-dioxa-13-azoniapentacyclo[11.8.0.0^2,10^.0^4,8^.0^15,20^]henicosa-1(13),2,4(8),9,14,16,18,20-octaene chloride, commonly known as berberine, was selected to be evaluated in a model of acute venous hypertension [57,58,59].

### 2.6. Effects of Ligation of the Veins on the Venous Pressure of the Left Hind Limb

Peripheral acute venous hypertension was induced in anesthetized rats by ligation of the right hind limb (RHL). After 15 min of stabilization, the mean arterial pressure (MAP), heart rate (HR), and venous pressure (VP) readings were taken. Figure 7 illustrates the comparison of VP values of the right hind limb (RHL) and left hind limb (LHL) before and after the ligation of the LHL veins. There were no significant differences in the baseline MAP and HR values before and after ligation (Table 5). In contrast, the ligation of four veins (superficial epigastric, proximal caudal, medial proximal geniculate, and popliteal) in the LHL increased the VP significantly both for rats that will be administered with berberine from a baseline of 11.42 mm Hg to 28.28 mm Hg (Figure 7A) and β-escin baseline of 11.4 mm Hg to 31.48 mm Hg (Figure 7B).

### 2.7. Effects of Berberine, and β-Escin on MAP, HR, and VP of the Anesthetized Rat

Successive injections of berberine (0.001, 0.003, 0.01, 0.03, 0.1, 0.3, 1 mg/kg, i.v.) and β-escin (0.56, 1, 17, 31, 56, and 10 mg/kg) were administered independently in anesthetized rats with venous ligation in the LHL. The last doses of berberine reduce the MAP with an increase, probably with a reflecting increase in HR (Figure 8A). In contrast, the administration of β-escin significantly reduces the HR and MAP with a higher dose than that employed in the experiments performed with berberine (Figure 8B). The administration of increased berberine doses resulted in a significant reduction in acute venous hypertension induced by the ligation of different veins in the LHL (Figure 9A). A similar effect but of lesser intensity was produced by the administration of increasing doses of β-escin (Figure 9B). The VP of the RHL used as a control does not show significant differences with the administration of berberine or β-escin. Figure 1 summarizes the steps followed in this work.

## 3. Discussion

Drug discovery from natural products has a long and successful history. Natural product research through computer-aided drug design techniques has been broadly successful, particularly when we explore natural products’ chemical space to identify bioactive compounds, with an emphasis on drug discovery [60]. In this study, we employed the application of chemoinformatics and network pharmacology techniques to identify compounds that potentially exhibit usefulness in VD. The selection of natural products as the basis of our investigation was motivated by the prevalence of natural-origin compounds called phlebotonics in VD treatment, which have strong associations with important signaling pathways [61,62,63,64,65,66,67,68].

In this study, we found 145 compounds reported with usefulness in the VD in three databases: MEDLINE, EMBASE, and Scopus. After clustering analysis, we discovered three groups with similar molecular descriptors. The main cluster of similar drugs involved in VD includes escin, crocin, echinomycin, cyclosporin A, amphotericin B, everolimus, rapamycin, chetomin, astragaloside IV, 20(R)-ginsenoside Rh2, epigallocatechin 3-gallate, troxerutin, O-(beta-hydroxyethyl)-rutoside, salvianolic acid B, keracyanin chloride, rutin, hidrosmine, diosmine, hesperidin, linarin, isorhoifolin, naringin, and betanin. The cluster that contained the most representative reference compounds was corroborated by the PCA analysis.

Our dataset shows a strong clustering tendency in flavonoid-related molecular descriptors. Flavonoids provide many health benefits, such as protecting against cardiovascular diseases, allergies, diabetes, inflammation, and oxidative damage [11].

Interestingly, all of them have different action mechanisms with potential usefulness in VD. For instance, escin is the active component of *Aesculus hippocastanum*, the horse chestnut, which has been used in traditional medicine for centuries and is currently used to treat hemorrhoids, varicose veins, hematoma, and venous congestion; this compound has shown anti-edematous, anti-inflammatory and venotonic activity. Its chemical structure contains a trisaccharide linked to the 3-OH residue, such as glucose, xylose, and galactose. Moreover, esterified domains in the C21 and C22 with an organic acid such as angelic, tiglinic, or acetic acid are also present [12]. Crocin, a water-soluble carotenoid, has been reported to demonstrate strong antioxidant activity against reactive oxygen species. Its mechanism is defined by the NF-κB pathway and NF-κBp65 translocation. This translocation inhibits the secretion of pro-inflammatory cytokines. Echinomycin is a cyclic peptide with antibacterial activity from the quinoxaline family produced by a strain of *Streptomyces echinatus*. This compound inhibits a factor involved in the signaling pathways in varicose veins, hypoxia-inducible factor-1 (HIF-1) [69]. Cyclosporine A is a cyclic non-ribosomal peptide of eleven amino acids. This peptide has been reported to have both immunosuppressive and mitochondrial inhibition activity. These characteristics have been given to Cyclosporine A to be used as a treatment for inflammatory pathologies, for instance, coronary artery disease (CAD), and acute myocardial infarction (AMI). Also, the pro-inflammatory genes IL-1β, IL-6, and tumor necrosis factor (TNF)-α have been targeted by this cyclic non-ribosomal peptide [70], which is also involved in VD [61]. Amphotericin B, a polyene antifungal with reported antibiotic activity, down-regulates the upregulation of inflammatory-related genes such as IL-6 and IL-8, reported in TNF-α-stimulated gingival epithelial cells with the involvement of p38 MAP kinase and ERK [71]. Both interleukins are targeted in the process of VD. Rapamycin and everolimus, which are mammalian targets of rapamycin (mTOR) inhibitors, have been reported to have significant anti-inflammatory activity. Their mechanism is comprised of avoiding the release of IL-8 and vascular endothelial growth factor (VEGF), allowing the preservation of the anti-inflammatory cytokine interleukin-1 receptor antagonist (IL-1RA). Pre-incubation with everolimus has shown a significative potent effect in TNF-α-treated neutrophils, which play a role in VD. This pre-incubation reduces the release of VEGF, IL-8, and IL-1RA [72]. Chetomin is an HIF-1α/p300 interaction inhibitor, a factor involved in VD [19]. The astragaloside IV (AS-IV), a 3-O-beta-D-xylopyranosyl-6-O-beta-D-glucopyranosylcycloastragenol extracted from the Chinese medical herb *Astragalus membranaceous* (Fisch), which has been reported to have in vivo anti-inflammatory activity, has been shown to inhibit the TNFalpha-induced specific mRNA levels for E-selectin and VCAM-1 [20]. 20(R)-ginsenoside Rh2, a minor stereoisomer of ginsenoside Rh2, has been reported to be an outstanding phlebotonic reference drug because of its several activities involved in VD such as matrix metalloproteinase inhibitory, anti-inflammatory, and anti-oxidative activity [73]. Epigallocatechin-3-gallate (EGCG) is a type of catechin found in green tea with reported anti-inflammatory and antioxidant properties [22], while troxerutin, is a flavonoid with radioprotective, antioxidant, and other diverse pharmacological activities [23]. O-(beta-hydroxyethyl)-rutosides (HR) is employed to treat chronic venous disease, as well as the indications of chronic venous insufficiency (VD), varicose veins, and deep venous disease [74]. Salvianolic acid B has been proposed to be an excellent drug candidate for the treatment and prevention of cardiovascular diseases [75]. Keracyanin chloride has been reported to reduce the expression of endothelial inflammatory antigens [26]. Rutin, hidrosmin, diosmin, hesperidin, and linarin have been used as pharmacological treatments for VD [6]. Linarin is a component of a micronized purified flavonoid fraction (MPFF) called Daflon. MPFF mainly contains diosmin (90%) and other active flavonoids like hesperidin, diosmetin, linarin, and isorhoifolin. It is used to treat inflammation and chronic venous insufficiency [76]. Naringenin is a flavonoid belonging to the flavanones subclass. It is widely distributed in several citrus fruits, bergamot, tomatoes, and other fruits. Several biological activities have been ascribed to this phytochemical, among them antioxidant and anti-inflammatory activity [34]. Betanin is widely found in red beet and is the most common beta cyanin pigment that acts as a stimulator of antioxidant defense mechanisms, has considerable free-radical scavenger activity, and is an anti-inflammatory agent [35].

Lipinski’s rule of five is commonly employed to assess the drug-likeness of a compound. Nevertheless, the rule does not extend to substrates of biological transporters or natural products [77]. According to research [77], 16% of approved oral drugs do not adhere to at least one of Lipinski’s criteria, while 6% fail to meet two or more. Because of this rationale, we opt to substitute Lipinski’s rule of five with the Quantitative Estimation of Druglikeness (QED) as the benchmark for assessing the drug-likeness of NP and SMSs [77]. This estimation enabled us to enhance our acquisition of NPs and SMSs that could potentially serve as drug candidates. Within this context, the QED displayed greater leniency in the selection criteria. QED allowed us to avoid bias and enrich our selection of NPs and SMSs, obtaining a more realistic and graded result. In reality, many approvals violate the criteria of this rule [77]. The molecular descriptors’ distribution showed resemblance in both datasets, even with the presence of molecules of different characteristics, encompassing both natural and synthetic products, in the phlebotonic dataset. Consequently, these data suggest that natural products could serve as a dependable resource for drug discovery.

Although there are many clustering methods, only some are used in practice; two were used in this study: hierarchical grouping and the *K-means*. They have been used successfully in fingerprint analysis [78]. Within the scope of this study, the utilization of fingerprint analysis facilitated the acquisition of 1846 molecules because they resemble reference molecules found in cluster 1. This finding suggests that the physicochemical characteristics of these 1846 compounds exhibit a higher level of structural similarity to reference compounds. Various clustering methods were utilized to validate these analyzes. The establishment of a compound–target network facilitated the identification of compounds with potential usefulness in VD based on their therapeutic targets. The collected data aided us in confirming that the similarity in chemical structure can lead to similar biological functions. This is clear as certain NPs exhibited targets associated with various pathways related to VD, including the inflammatory response pathway, cellular response to hypoxia, PI3K-AKT-NFKB pathway, ion channel transport, Il-18 signaling pathway, VEGF-VEGFR-2, senescence-associated secretory phenotype (SASP), oxidative stress, and intrinsic pathways for apoptosis. However, few SMSs from the AC discovery database had reported targets in PubChem and none were related to VD pathways. Hence, it is crucial to conduct additional bioassays to identify more drug targets and document them in these databases. It is crucial to consider compounds that may not have direct associations with VD targets to prevent them from being disregarded, as some of these compounds might display phlebotonic activity because of structural similarities with compounds identified in cluster one of phlebotonics.

Through this study, compounds with a multi-target capacity were identified, which had been previously selected based on fingerprints. Thus, the development of multi-target drugs is essential as a promising strategy to address complex, multifactorial disorders [79], including VD [61]. Through the analysis of the structure and functionality of multi-target molecules, this study has identified three compounds that exhibit an enhanced likelihood of selectively acting on VD. These new molecules open a pre-clinic study field of great interest considering the knowledge regarding the NPs that must be confirmed in preclinical studies.

Traditional Chinese medicine has been widely used for the treatment of various diseases, obtaining great relevance within clinical applications. VDs that affect the vasculature of the heart, cerebrovascular disease, atherosclerosis, and diabetic complications have harmed the quality of life of patients and present an increase in the burden of health care services. Berberine, an isoquinoline alkaloid from *Rhizoma coptidis*, has wide application in traditional Chinese medicine because of its antibacterial and anti-inflammatory properties. With this background, a greater number of studies have been generated that have allowed us to identify several cellular and molecular targets for berberine, indicating its potential as an alternative therapeutic strategy for vascular diseases, in addition to providing novel evidence supporting the potential therapeutic use of berberine to combat VD. Currently, berberine displays remarkable anti-inflammatory, antioxidant, antiapoptotic, and anti-autophagic activity through the regulation of multiple signaling pathways, including AMP-activated protein kinase (AMPK), nuclear factor κB (NF-κB), mitogen-regulatory Silent Information System Activated Protein Kinase 1 (SIRT-1), Hypoxia-Inducible Factor 1α (HIF-1α), Phosphoinositide Vascular Endothelial Growth Factor 3-Kinase (PI3K), Protein Kinase B (Akt), Janus Kinase 2 (JAK-2), Ca^2+^ channels and endoplasmic reticulum stress, in addition to modulating Na^+^, Ca^2+^ concentration and lipid metabolism in vascular smooth muscle cells [47]. All these actions increase the possibility of obtaining therapeutic activity in venous activity because of its similarity with the structure of tested phlebotonic drugs and its multi-target ability as a protector in various vascular according to recent in vitro and in vivo experimental reports.

In our study, berberine was utilized due to its potential venous antihypertensive effect and its ability to protect vascular endothelial cells, enhance vascular remodeling, and inhibit inflammation, oxidative stress, autophagy, and apoptosis. The results of our experimental model of peripheral venous hypertension confirm a significant venous antihypertensive effect. However, the evaluation conducted so far focused on acute venous hypertension, necessitating an assessment of a chronic hypertension model to determine the efficacy of berberine in mitigating microcirculatory changes, safeguarding endothelial cells, reducing inflammation, and preventing ischemia by enhancing endothelial function.

It is widely recognized that phlebotonic activity extends beyond the mere elevation of venous wall tone. In the modern context, veno-active drugs should possess the ability to stimulate lymphatic drainage and enhance microcirculation [80], employing various mechanisms [81,82,83,84]. Endothelial dysfunction holds significance within the venous system as it may lead to a decrease in venodilation, resulting in an increase in venous tone and a decrease in venous compliance. This, in turn, enhances the mean circulatory filling pressure, leading to the maintenance or alteration of cardiac workload and contributing to the development of cardiovascular diseases. Changes in vein function may manifest early on, even before the onset of these diseases. However, is not yet fully understood if the venous endothelium dysfunction is involved in these alterations, so further studies are required [85]. This hypothesis may support our idea that berberine could be a more efficient phlebotonic than β-escin, because berberine generates vasorelaxation, avoiding endothelial damage and thus preventing the progression of VD. A similar process has been reported during the treatment with the phlebotonic rutin [86]. Our discovery supports the notion that berberine stimulates vasorelaxation mediated by the endothelium and enhances vasodilation in VSMCs by partially reducing oxidative stress [87], so we expect that these positive effects may present in the venous environment.

The effects of vasorelaxation may be involved because the targets of berberine found in our compound–target network were the KCNH2 and NFE2L2 genes as the most relevant. KCNH2 is a voltage-activated potassium channel found in cardiac muscle, nerve cells, and microglia. This gene encodes the pore-forming subunit of a rapidly activating-delayed rectifier potassium channel that plays an essential role in the final repolarization [88]. NFE2L2 is a transcription factor that shapes the antioxidant response in driving cancer progression, metastasis, and resistance to therapy [89].

The use of virtual screening and network pharmacology allowed us, for the first time, to find that berberine produces venous antihypertension activity. Our finding, coupled with the background of berberine activities, makes it a better candidate with usefulness in VD because it is a natural product with therapeutic potential in vascular diseases [47]. In addition, it has been shown to improve other conditions that are associated with VD, such as diabetes mellitus [47].

These experimental results belong only to an evaluation of one of many other candidates found in this work, which will be tested in a model of chronic venous hypertension that is currently under development.

## 4. Materials and Methods

To achieve the proposed aims, several were created:

### 4.1. In Silico Phase

#### 4.1.1. Creation of a Reference Compound Dataset

##### Collection of Useful Compounds in VD

We created a dataset by reviewing literature from MEDLINE, EMBASE, and Scopus up to March 2023, focusing on useful compounds in VD. The search descriptors in this work were “venous disease”, “phlebotonics”, “venotonics”, “venous insufficiency treatment”, “venous disease treatment”, “chronic venous disease”, and “pathways”.

##### Selection of Reference Compounds from the Dataset Based on Hierarchical Analysis and Chemical Space Analysis

The molecular structure of each compound found in the literature (reference compounds dataset) was translated into a Simplified Molecule Input Line Entry System (*SMILES*) obtained from PubChem (https://pubchem.ncbi.nlm.nih.gov/) [90] (accessed on 3 April 2023). Subsequently, the SwissADME server (http://www.swissadme.ch) [91] (accessed on 10 April 2023) and Osiris Data Warrior software V5.2.1 were used to analyze the physicochemical properties and molecular descriptors. Total Molweight, Molweight, Monoisotopic Mass, cLogP, cLogS, H-Acceptors, H-Donors, Total Surface Area, Relative PSA, Polar Surface Area, Druglikeness, Shape Index, Molecular Flexibility, Molecular Complexity, Fragments, Non-H Atoms, Non-C/H Atoms, Electronegative Atoms, Stereo Centers, Rotatable Bonds, Rings Closures, Aromatic Atoms, sp3-Atoms, Symmetric atoms, Small Rings, Carbo-Rings, Hetero-Rings, Saturated Rings, Non-Aromatic Rings, Aromatic Rings, #Heavy atoms, #Aromatic heavy atoms, Fraction Csp3, MR, TPSA, log Kp (cm/s), Lipinski #violations, Ghose #violations, Veber #violations, Egan #violations, Muegge #violations, Bioavailability Score, and Synthetic Accessibility were the molecular descriptors for this work [92].

By using the K-means algorithm and distance matrix, similarities in the molecular descriptors of the reference compounds were identified. This analysis was performed with the *complexheatmap* package [93] and R-studio version 3.4. PCA was used to analyze the chemical space and data distribution to validate these results. PCA is defined as an orthogonal linear transformation technique that can transform the data into a new coordinate system. We utilized a two-dimensional system in our analysis [94]. Briefly, the variance of the data maximized on the first coordinate was called the first principal component. The second variance was maximized on the second coordinate; we used the *factoextra* package with R-Studio version 3.4.

#### 4.1.2. Drug-Likeness of NPs and SMSs from BIOFACQUIM, Inflamnat, NuBBE, and AC Discovery Databases Was Determined Using the QED Index

##### Data Collection of NPs and SMSs

To obtain compounds with potential usefulness in VD, in this work, we used different databases that contained both NPs and SMSs. NPs were obtained from databases like InflamNat, BIOFACQUIM, NuBBE, and AC Discovery. SMS were only obtained from the AC Discovery database.

We analyzed only the molecules that were available on that server.

##### Determination of the Drug-Likeness of NP and SMSs

To determine the drug-likeness of NP and SMS compounds, a drug-likeness estimator based on the Quantitative Estimation of Drug-likeness (QED) was used. This one is part of RDKit, a piece of open-source cheminformatics software (version 2023.03) [95] (https://www.rdkit.org/docs/source/rdkit.Chem.QED.html (accessed on 10 April 2023)). The QED is represented as an integrative score to evaluate the compounds’ preference to be considered a hit [77]. This method quantifies the drug-likeness, considering the following molecular descriptors: molecular weight (Da), Ghose–Crippen–Viswanadhan octanol–water partition coefficient (AlogP), the number of H-acceptors (HBA), the number of H-donors (HBD), the number of rotatable bonds, total polar surface area (TPSA), and aromatic bound account. We analyzed the molecular descriptors of the reference compounds, NPs, and SMSs to identify similar patterns in the datasets. A histogram and box plot were accomplished to represent graphically this distribution. The score of QED uses the molecular descriptors mentioned above. We selected the NPs and SMSs with high QED scores (over 0.5).

##### Filtering of the NP and SMS Dataset

To eliminate NPs and SMSs with potential non-desirable effects, we identified the functions related to tumorigenesis, reproduction, and other effects [96]. Osiris Datawarrior software V5.2.1 was employed to determine these potential non-desirable effects. Because of the large number of drug-like molecules available, we used PCA to select molecules that resembled phlebotonics in their chemical space. We utilized both two- and three-dimensional systems in our analysis comparing the NPs and SMSs versus reference compounds [94,97]. We used Osiris Datawarrior software V5.2.1 for the determination of PCA. Only the molecules with similar chemical space to reference compounds were considered for the next step of the work.

#### 4.1.3. The Selection of Compounds with Potential Usefulness in VD Based on Comparison of the Fingerprints of the Drug-like NPs, SMSs, and Reference Compounds

##### Fingerprints of Drug-like NPs, SMSs, and Reference Compounds

We used fingerprint analysis to identify molecules similar to the reference compounds present in our drug-like NPs in the SMS dataset. Molecular fingerprints are one of the most common representations of chemical structures in chemoinformatics. They represent chemical information in any chemical entity through binary vectors. Commonly, the binary fingerprint is the representation of a chemical structure. The vector’s positions show the presence (1) or absence (0) of predetermined features in the fingerprint design [98].They are essential cheminformatic tools for virtual screening and mapping chemical space [99]. This method was supplemented with clustering of chemical compounds by the similarity of their molecular fingerprints to identify similar structures in an extensive set of similar data, as previously reported [78]. Reference compounds were compared to drug-like molecules using a fingerprint analysis. To achieve this structural comparison, we used *ChemmineR* and *rcdk* in RStudio version 3.4 [78]. Because the molecular structures from all datasets were aromatic compounds, the “extended” method is preferred. An extended number of fingerprints with a default length of 1024 (number of bits) were used. The preferred analysis gave us a higher level of detail in the structure analysis compared to the MACCS-type fingerprints. By using the Tanimoto coefficient, we compared the similarity of NPs, SMSs, and reference compounds through cluster analysis. Later, the molecules were classified into three categories based on their distances using Ward’s clustering method, a widely used algorithm in drug discovery. To confirm the clustering results, the Dunn index and the silhouette coefficient were used [78]. The cluster, which had a reference compound, was employed to search for compounds with potential usefulness in VD and generate a compound–target network.

#### 4.1.4. Selection of the Best Compounds with Potential Usefulness in VD through Their Multi-Target Capacity

##### Target Search of the Compounds Obtained Using Fingerprint Analysis

The pharmacological networks were constructed by searching for the reported targets of each drug-like compound in the PubChem server. Only drug-like molecules with reported targets were considered for this phase of the study.

##### Compound–Target Network Generation

We used network pharmacology to select compounds with potential usefulness that interact with VD networks. By linking drug-like compounds and their reported targets with genes from VD-related pathways in Wikipathways, this network was formed. The pathways used in this analysis were the inflammatory response pathway, cellular response to hypoxia, PI3K-AKT-NFKB pathway, ion channel transport, IL-18 signaling pathway, VEGF-VEGFR-2, senescence-associated secretory phenotype (SASP), oxidative stress, intrinsic pathways for apoptosis, and angiogenesis [61,62,63,64,65,66,67,68]. For more detail on the targets, pharmacological functions, and genes data from the VD network, see the Appendix A.

Cytoscape software version 3.8.2 was used to construct the compound–target network [100]. Only multi-target compounds participating in over three pathways were considered as compounds with potential usefulness in VD.

##### Selection of the NPs and SMSs through Their Availability, Accessibility(cost), and Ease of Administration

The best candidates with potential usefulness in VD obtained using network pharmacology were evaluated in an in vivo model. We started with various criteria based on their availability, accessibility, and administration. The information was collected from the literature and commercial sites. The Figure 1 shows a summary of the strategy followed in this work.

### 4.2. Experimental Phase

#### 4.2.1. Materials

Berberine chloride (Sigma Aldrich Cat. B3251-10G) (St. Louis, MO, USA) was dissolved in normal saline-dimethyl sulfoxide (DMSO) (9:1). β-escin (Sigma Aldrich Cat. E-1378 10G) was dissolved in phosphates buffer to a pH of 7. Urethane (Sigma Aldrich Cat. U-2500 500G)/chloralose (Sigma Aldrich Cat. C-0128 500G) (1000/100 mg) was dissolved in water.

#### 4.2.2. Animals

Male Wistar rats weighing 300–350 g were used for all experiments. The animals were maintained at room temperature (21–23 °C) on a 12 h light:12 h dark cycle and fed with pellets food (5001 Rodent Laboratory Chow) ad libitum. Their care was in line with Mexican standards [101,102] and international official guidelines [103]. The procedures were approved by the Research Ethics Committee of the Faculty of Medicine, UNAM (Project No. 008-CIC-2022).

#### 4.2.3. Register of PAM, HR, and VP in Anesthetized Rats

Wistar rats were anesthetized with urethane/chloralose (1000/100 mg/kg i.p. 100 g/0.2 mL). Using extreme care to prevent skin irritation, the limbs and throat were shaved. Through vertical incisions, the veins of the thigh, such as the superficial epigastric, proximal caudal, medial proximal geniculate, and popliteal [104], were surgically exposed and subsequently dissected. These veins of the left hind limb were ligated with microvascular clips (11 mm) (George. Tiemann Co., Long Island City, New York, NY, USA). The veins of the right hind limb were not ligated (sham control group).

In all animals, PE 50 cannulas (BD IntramedicTM, Sparks, MD, USA) were inserted into the right carotid artery, left jugular vein, and right and left femoral veins to measure medium arterial pressure, heart rate, and venous pressure in both hind limbs, drug administration and recording of the venous pressure, respectively. In the trachea, a PE 240 cannula was inserted to facilitate spontaneous breathing. Model P231D pressure transducers (Gould-Statham Instruments, Oxnard, CA, USA) were connected to the cannulas inserted in the carotid artery and femoral vein to record MAP, HR, and PV [105,106]. The signal from the transducer was electronically damped and recorded using a Model 79 Grass polygraph (Grass Instrument, Quincy, MA, USA). HR was recorded through another channel from the polygraph with a Grass 7P4 tachograph triggered by the pulse waves from the unfiltered transducer signal, and the pressure change was transmitted to The software Lab-ViewTM 21 SP1 (National Instruments, Austin, TX, USA) via an interface NI USB-6009 multifunction DAQ (National Instruments, USA). The animals’ temperature was regulated at 38 °C through the utilization of a heating table connected to a regulation unit. All experiments used an electronic thermometer to regulate the temperature probe control of the heating table. The animals were stabilized for 15 min after the insertion of the cannula.

#### 4.2.4. Dose–Response Curves of the Cardiovascular System to Berberine, and β-Escin

After 15 min of stabilization, MAP, HR, and VP readings were taken. Successive injections independent of berberine (0.001, 0.003, 0.01, 0.03, 0.1, 0.3, 1 mg/kg, i.v., and β-escin (0.56, 1, 17, 31, 56 and 10 mg/Kg) were administered. A time interval separated each bolus injection to allow recovery of the stability of MAP, HR, and PV in both hind limbs. The group comprised eight animals receiving berberine and six receiving β-escin treatment. The obtained results were used to construct dose–response curves.

For more details, see the section before the Introduction. All the acronyms/initialisms/abbreviations are deciphered within it.

## 5. Conclusions

Through our analysis of chemical structures and compound–target networks, we have identified drug-like compounds with potential usefulness in VD. We identified 18 compounds with potential usefulness in VD. Only the following compounds have a previously reported method for experimental use: 7-hydroxyflavone, acacetin, berberine chloride, biochanin A, chrysin, formononetin, and sulfuretin. Berberine, our chosen candidate, significantly reduced peripheral acute venous hypertension in anesthetized rats. This effect shows the start of confirmation to our approach for discovering new drugs using virtual screening and network pharmacology. Additional preclinical experiments are needed to determine whether berberine is effective for treating chronic peripheral venous hypertension.

Our results show that testing the most promising candidates in future experiments could lead to the discovery of new VD drugs. VD is a complex illness for which there is a lack of experimental models.

## Data Availability

The data presented in this study are available in the Appendix A.

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
