# Peer review of "Discovering Potential Compounds for Venous Disease Treatment through Virtual Screening and Network Pharmacology Approach"

_molecules, 2023, doi:10.3390/molecules28247937_

Round 1

Reviewer 1 Report

Comments and Suggestions for Authors

The manuscript deals with very interesting topic, but from time to time it is hard to understand the purpose of many sentences. 

The Results section must be rewritten to introduce readers with the purpose of many performed steps. The names of the compounds must be according to IUPAC rules. The abbreviations must be defined. Why did you use PCA? Why did not use MANOVA? Why did not use LDA? You mentioned fingerprint analysis, but you did not give any sound explanation except through figure. 

Comments on the Quality of English Language

The language must be improved. 

Author Response

We appreciate your observation, which is considerably correct and very pertinent. The authors apologize for the redaction of the manuscript that made it hard to read. In our new version, we tried to be more clear in the redaction of the sentences both in methods and results.

Reviewer 2 Report

Comments and Suggestions for Authors

Thank you for your work!

The authors wrote very extensive yet legible original article on different approaches for drug discovery to treat venous disease. They revealed the set of potential compounds and expediently chose one of them – berberine – to test it in the experimental models. Somehow the description of the second experimental model using isolated venous vessels, as well as the results on it, are missing in the text. Please pay attention to the Supplementary Materials that need to be added, make them clear and attractive, and resolve the issue with a missing part on isolated venous vessels (either exclude it completely, which will not worsen the article, or include in full). In general, this work seems to be very promising and builds a platform for future research thereafter. The manuscript deserves to be published after taking into account all the remarks and making corrections.  

The comments include requirements and recommendations.

1) ‘phlebotonics compounds’ (line 16) – it’s better to replace it with ‘phlebotonic compounds’

2) In the Abstract (line 22) ‘Because of their descriptors and multi-target capacity, 19 compounds were identified as potential…’: does the adj. ‘their’ relates to those 19 compounds (which is more likely) or to 84 molecules interacted with the VD network (line 21)?

3) ‘This method has proven effective’ (line 64) needs to be replaced with ‘This method has proven to be effective’.

4) ‘The results can be observed in Figure 1 and Supplementary Figure 1’ (lines 75-76), ‘Supplementary Figure 2 shows’ (line 137), ‘See the Supplementary materials for more information’ ‘(lines 554-555)’: those phrases refer to the Supplementary Materials that are not attached to the manuscript except “molecules-2719272-supplementary.xlsx” file containing the tables, which makes it difficult for a reviewer to evaluate this work. Perhaps the files were somehow forgotten to submit along with the manuscript. In ‘Supplementary Materials: The following supporting information can be downloaded at: www.mdpi.com/xxx/s1, Figure S1: title; Table S1: title; Video S1: title (lines 638-639) there are no titles for Figures and Tables at all. This issue needs to be resolved.

5) Please mention that Appendix A contains all the acronyms/initialisms/abbreviations (such as CADD, NPs, SMSs, …) deciphered within it.

6) ‘2.6. Effects of ligation of the veins on the PV of the LHL’ (line 167):  PV should be replaced with VP (venous pressure), I suppose.

7) In the Abstract (lines 24-25), Introduction (lines 67-68), and Results (lines 165-166) sections, the authors wrote that despite using a rat model of acute venous hypertension they also performed their experiments on isolated inferior vena cava. But the last paragraphs of the Results section are ‘2.6. Effects of ligation of the veins on the PV of the LHL’ (lines 167-175) (where the rat model used is proved to be valid) and ‘2.7. Effects of Berberine, and β-escin on MAP, HR, and VP of the anesthetized rat’ (lines 176-187) (where the effects of Berberine and a phlebotonic reference β-escin on certain parameters were investigated using the chosen animal model) followed by the Scheme 1 where the last step (phase 5) shown is ‘In vivo model of acute venous Hypertension of rat’. In fact, there are no any results on isolated inferior vena cava presented in the manuscript: neither in the Results, Discussion and Materials and Methods section (the last paragraph is ‘4.2.4. Dose-response curves of the cardiovascular system to berberine, and β-escin’ (lines 603-609)). If the authors did not do the experiments on isolated venous vessels (or maybe just planned to do) – that is totally fine, the work done is enough to be publish. But the authors definitely need to resolve the issues with the text.

8) The Conclusions section (lines 610-637) looks to me too long, it’s better to shorten it and make more concise, but that is optional. 

Author Response

(The authors gave the same response as above.)

Round 2

Reviewer 1 Report

Comments and Suggestions for Authors

The manuscript was improved, but there are still points that need clarifications.

The language should be improved because there are still spelling and grammar mistakes.

Lines 96 and 97: What do you mean by one-dimensional (1-D) and two-dimensional properties (2-D)?

O- and (R) must be written in italic, and all others like (E)-,... according to IUPAC rules.

"Phase 2", "Phase 3" and similar labels should be deleted. 

In Table 1, besides IUPAC names, the labelling must be in accordance with IUPAC rules. The same suggestion is valid for Table 3.

Lines 341 and 373: Please make proper sentences.

Lines 457-462: It is very hard to understand. This part must be rewritten.

Line 463: It is completely insane sentence. Please remove it.

Line 505: Please give a proper sentence. It is not clear what you wanted to say.

Lines 585-586: I think that this sentence is not necessary to be here because it was already mentioned before.

Line 594: This is not a fashion magazine. Therefore, please delete: "a popular algorithm in drug discovery".

Lines 611-612: Please rewrite the sentence to be scientifically sound.

I am still not satisfied with the performed PCA analysis and arbitrary grouping. Therefore, I suggest you to perform MANOVA, and later LDA.

All your cluster plots of all compounds (C) in the Supplementary Material look the same. I do not believe your data. Please check all (C). Also (B) in all figures must be readable.  

Comments on the Quality of English Language

The language must be improved. 

Author Response

Dear reviewer:

In this second round, we answer the major revisions requested. We have improved and enriched the present manuscript. This new version of the manuscript contains complementary information both in the text and in the supplementary figures that have been added to clarify our work. Also, we included additional files with statistical tests and clustering information which can be downloaded  from this link (https://drive.google.com/file/d/1eOVHcnzDrPAzlh7Yg5D_2fW3LgyWj6BU/view?usp=sharing) . These files showed why the PCA was employed as a complementary and selection test in our research. 

Best wishes,

Dr. Magos-Guerrero
